# Antitumor Effect of Korean Red Ginseng through Blockade of PD-1/PD-L1 Interaction in a Humanized PD-L1 Knock-In MC38 Cancer Mouse Model

**DOI:** 10.3390/ijms24031894

**Published:** 2023-01-18

**Authors:** Eun-Ji Lee, Ju-Hye Yang, Hye Jin Yang, Chong-Kwan Cho, Jang-Gi Choi, Hwan-Suck Chung

**Affiliations:** 1Korean Medicine Application Center, Korea Institute of Oriental Medicine, Daegu 41062, Republic of Korea; 2East-West Cancer Center, Daejeon Korean Medicine Hospital of Daejeon University, Daejeon 35235, Republic of Korea

**Keywords:** immune checkpoint, PD-1/PD-L1 inhibitor, Korean Red Ginseng, cancer immunology, humanized PD-1 mice

## Abstract

Blocking immune checkpoints, programmed death-1 (PD-1) and its ligand PD-L1, has proven a promising anticancer strategy for enhancing cytotoxic T cell activity. Although we previously demonstrated that ginsenoside Rg3, Rh2, and compound K block the interaction of PD-1 and PD-L1, the antitumor effect through blockade of this interaction by Korean Red Ginseng alone is unknown. Therefore, we determined the effects of Korean Red Ginseng extract (RGE) on the PD-1/PD-L1 interaction and its antitumor effects using a humanized PD-1/PD-L1-expressing colorectal cancer (CRC) mouse model. RGE significantly blocked the interaction between human PD-1 and PD-L1 in a competitive ELISA. The CD8^+^ T cell-mediated tumor cell killing effect of RGE was evaluated using murine hPD-L1-expressing MC38 cells and tumor-infiltrating hPD-1-expressing CD8^+^ T cells isolated from hPD-L1 MC38 tumor-bearing hPD-1 mice. RGE also reduced the survival of hPD-L1 MC38 cells in a cell co-culture system using tumor-infiltrating CD8^+^ T cells as effector cells combined with hPD-L1 MC38 target cells. RGE or Keytruda (positive control) treatment markedly suppressed the growth of hPD-L1 MC38 allograft tumors, increased CD8^+^ T cell infiltration into tumors, and enhanced the production of Granzyme B. RGE exhibits anticancer effects through the PD-1/PD-L1 blockade, which warrants its further development as an immunotherapy.

## 1. Introduction

During cancer progression, tumors evolve and develop a variety of mechanisms to evade immune surveillance and suppress the antitumor immune response. The basic mechanism of tumor immune evasion involves immune checkpoint pathways. Immune checkpoints have been considered antitumor therapeutic targets as they regulate immune evasion in cancer cells [1].

Immune checkpoint inhibitors (ICIs) have appeared as innovative and effective immunotherapies for cancer. ICIs assist the patient’s immune system in mounting an antitumor response. Among the various ICIs, programmed cell death (PD-1)/programmed cell death ligand 1 (PD-L1) blockers exhibit remarkable therapeutic effects against various cancers [2,3]. Currently, the Food and Drug Administration (FDA) has approved the PD-1/PD-L1 blockers atezolizumab, avelumab, cemiplimab, durvalumab, nivolumab, and pembrolizumab for cancer treatment [4]. However, side effects related to resistance and immunity have been reported, and treatment is expensive. Therefore, the development of an effective and safe PD-1/PD-L1 blocker is required, and many studies are currently underway [5].

Korean Red Ginseng (KRG) is processed by steaming or drying ginseng, and contains a high concentration (60–70%) of carbohydrates including starch. In addition, it contains ginsenosides, which are various physiologically active substances that do not exist in other plants, nitrogen-containing compounds, such as proteins, peptides, and alkaloids, and fat-soluble polyacetylenes, polysaccharides, and flavonoids [6]. In KRG, there are changes in the type and concentration of ginsenoside, a compound unique to ginseng, in the process of processing red ginseng, and there are physicochemical changes in polysaccharide, the most abundant compound in Korean red ginseng. In particular, it has a low moisture content of less than 15.5% and is widely used as a health functional food because it can be stored for a long time by gelatinizing starch.

KRG is a traditional herbal medicine that has been used in East Asia to treat various ailments, and is commonly used as an herbal medicine to treat allergies and inflammatory and cancer diseases [7]. Numerous studies have reported that KRG exhibits various biological activities, such as immunity enhancement [8,9,10], antiinflammation [11,12], antifatigue [13,14,15,16], antiobesity [17,18], antioxidant [19,20], menopausal relief [19,21], and anticancer effects [10,22,23,24]. Moreover, KRG inhibited endocrine disrupting chemical-induced endometriosis in a mouse model [25] and induced both extrinsic and intrinsic apoptotic pathways in breast cancer cells and non-malignant breast cells [26]. It has been reported that the non-saponin fraction of KRG regulates immune homeostasis in old mice [27], improves sarcopenia by regulating chronic inflammation and oxidative stress, and suppresses obesity and dyslipidemia in castrated mice fed a high-fat diet [28]. In addition, fatigue recovery effects through the control of 5-HT and corticosterone have been reported in a sleep-deprived mouse model [15]. Furthermore, KRG reduces the production of allergic and inflammatory cytokines and inhibits trans-epidermal water loss, and synergistic effects have been reported to inhibit atopic dermatitis when used in combination with probiotics [29]. However, the immuno-oncologic effects of KRG via immune checkpoint remains largely undefined.

In a previous study, we found that Rg3, Rh2, and compound K exhibited the potential to disrupt the PD-1/PD-L1 interaction in competitive ELISA and protein-ligand docking simulations [30]. Although Rg3, Rh2, and compound K are major components of RGE, RGE is primarily used as a health supplement rather than as a single compound, such as Rg3, Rh2, or compound K. The effects of RGE on the PD-1/PD-L1 interaction and immune system have not been thoroughly examined. In the present study, we determined the effect of RGE alone on the PD-1/PD-L1 interaction and its antitumor effects against a humanized PD-1/PD-L1-expressing CRC mouse model.

## 2. Results

### 2.1. RGE Blockade of Targeting Human PD-1/PD-L1

To evaluate the inhibitory effect of RGE on the interaction of human PD-1 (hPD-1) and human PD-L1 (hPD-L1), we conducted a competitive ELISA targeting hPD-1/hPD-L1. RGE exhibited strong inhibitory activity against the hPD-1/hPD-L1 interaction with a 50% inhibitory concentration (IC_50_) value of 176.2 μg/mL (Figure 1A). The IC_50_ for the neutralized antibody as a positive control for hPD-1 was 2.864 μg/mL and 4.103 μg/mL for hPD-L1 (Figure 1B,C). These data show that RGE is a potent immune checkpoint inhibitor targeting hPD-1/hPD-L1 in vitro.

### 2.2. RGE Enhances Tumor-Infiltrating CD8^+^ Cell-Mediated CRC Cell Killing

The inhibition of the hPD-1/hPD-L1 protein interaction by RGE prompted us to determine how RGE affects hPD-1-expressing CD8^+^ T cell-mediated, hPD-L1-expressing CRC cell killing in a co-culture model. Therefore, we established CD8^+^ T cell-mediated CRC cell killing assays using murine hPD-L1-expressing MC38 cells and tumor-infiltrating hPD-1-expressing CD8^+^ T cell isolated from hPD-L1 MC38 tumor-bearing hPD-1 mice. To confirm whether RGE affects the viability of hPD-L1 MC38 cells, the cells were treated with diverse concentrations of RGE for 72 h. RGE exhibited a cytotoxicity on the hPD-L1 MC38 cells at a concentration of 1000 μg/mL (Figure 2A), whereas the IC_50_ of RGE on the hPD-L1 MC38 cells was 656.6 μg/mL. RGE had no cytotoxic effect on tumor-infiltrating hPD-1-expressing CD8^+^ T cells up to a concentration of 300 μg/mL (Figure 2B). In addition, RGE treatment of CD8^+^ T cells increased the release of IL-2, suggesting that CD8^+^ T cells are activated by RGE (Figure 2C).

Considering the non-cytotoxic concentrations (300 μg/mL) of RGE on hPD-L1 MC38 and hPD-1 CD8^+^ T cells, a co-culture cell system was established using tumor-infiltrating CD8^+^ T cell as effector cells in conjunction with hPD-L1 MC38 cells as target cells to elucidate the CD8^+^ T cell-mediated antitumor effect of RGE. As shown in Figure 2D, the addition of tumor-infiltrated CD8^+^ T cells at 5-fold excess of the hPD-L1 MC38 cell number, RGE treatment effectively inhibited the amount of surviving hPD-L1 MC38 cells. The cytotoxicity of hPD-L1 MC38 cells co-cultured with tumor-infiltrated CD8^+^ T cells was increasingly high in a concentration-dependent manner (Figure 2E). Moreover, RGE treatment apparently activated cytotoxic CD8^+^ T cells, which are key mediators of cancer cell killing, resulting in the elevated release of GrB (Figure 2F). The results of the CD8^+^ T cell-mediated MC38 cell killing assay indicates that RGE efficiently enhanced the activation of tumor-infiltrating hPD-1-expressing CD8^+^ T cell immune function through inhibition of the hPD-1/hPD-L1 interaction.

Upregulation of PD-L1 in cancer cells is a way to evade immune surveillance. It is reported that the PD-L1 antibody binds to PD-L1 on the outer membrane then internalize the PD-L1 [31]. In addition, the PD-L1 is degraded in proteasome or lysosomes. RGE inhibited PD-L1 expression in MDA-MB-231 cells, the breast cancer cell line, which highly express PD-L1 (Appendix A).

### 2.3. Antitumor Effect of RGE in a Humanized PD-1/PD-L1 Knock-In Tumor Model

We demonstrated RGE suppression of hPD-L1 MC38 tumor growth in vivo by activated CD8^+^ T cells through inhibition of the hPD-1/hPD-L1 interaction. Next, we implanted hPD-L1-expressing MC38 cells in humanized PD-1 knock-in mice to evaluate the antitumor activity of RGE. Once the tumor volume reached 100 mm^3^ (day 14), the mice were administered oral RGE at a dose of 100 or 300 mg/kg every day, whereas Keytruda (hPD-1 antibody) was administered intraperitoneally at dose of 2.5 mg/kg twice a week. RGE 300 mg/kg and Keytruda treatment delayed tumor progression compared with the vehicle group (Figure 3A). RGE or Keytruda treatment markedly suppressed the growth of hPD-L1 MC38 allograft tumors, exhibiting a stronger inhibition compared with the vehicle group as monitored by decreased tumor weight and volume (Figure 3B,C). Tumor suppression rate in the RGE 300 mg/kg or Keytruda group was 60% and 70% on day 16, respectively (Figure 3D). The body weight of the mice treated with RGE or Keytruda showed no significant change compared with the vehicle group throughout the experiments (Figure 3E). These results indicate that RGE effectively suppresses tumor growth of hPD-L1 MC38 allograft tumors in hPD-1 mice without affecting body weight.

### 2.4. RGE Activates Tumor-Infiltrating CD8^+^ Cells in hPD-1/hPD-L1 MC38 Tumor Tissues

Considering the suppression of hPD-L1 MC38 tumor growth by RGE in in vitro, in vivo, and co-culture cell models, and the activation of tumor-infiltrating CD8^+^ T cells by RGE treatment, we examined CD8^+^ T cell infiltration into hPD-L1 MC38 tumor tissues following RGE or Keytruda treatment. As expected, the infiltration of CD8^+^ T cells and CD4^+^ T cells into tumors was markedly increased in the group treated with RGE 300 mg/kg or Keytruda (Figure 4A,B,D). In addition, intratumoral infiltrating cytotoxic CD8^+^ T cells produced more GrB in the group treated with RGE 300 mg/kg or Keytruda (Figure 4A−C). Taken together, our findings indicate that RGE enhances the abundance of CD8^+^ T cell infiltration on hPD-L1 MC38 tumors and showed greater antitumor activity in the hPD-1/hPD-L1 knock-in MC38 CRC tumor allograft mouse model.

### 2.5. Identification of Ginsenoside Rg3 of RGE by HPLC-DAD Analysis

The content of ginsenoside was greatly increased by RGE; thus, ginsenoside Rg3 was selected as a standard for analysis. As shown in Figure 5, ginsenoside Rg3 was well-separated and demonstrated good resolution, while minimizing interference from other analytes. Based on a comparison with the chromatogram, the retention time (tR; 30.25 min of standard, 30.24 min of RGM) and UV spectrum (203 nm) of the standard and ginsenoside Rg3 in RGE were detected.

### 2.6. Validation of the Analytical HPLC Method

The calibration curve for ginsenoside Rg3 was established by plotting the peak area versus the concentration. The linear correlation coefficient (R^2^) for the calibration curve was greater than 0.9999 (Table 1). Based on this curve, the amount of ginsenoside Rg3 was determined to be 6.0423 ± 0.0228 mg/mL.

## 3. Discussion

The interaction of PD-1/PD-L1 via extracellular domains is a well-known immune checkpoint associated with the escape of cancer cells from immune-mediated destruction by inhibiting the anticancer effect of T cells [32]. Blocking the PD-1/PD-L1 interaction is an important anticancer target and the immunotherapeutic effect is higher in cancer cells overexpressing PD-L1 [33]. CRC with a microsatellite instability-high (MSI-H)/deficient DNA mismatch repair (dMMR) phenotype contains a high tumor mutational burden and tumor-infiltrating lymphocyte (TIL) activation. This indicates the clinical significance of TILs in MSI-H/dMMR CRC patients, suggesting that they may respond to ICIs over a long time period [34]. Treatment of MSI-H/dMMR CRC targeting PD-1 or PD-L1 inhibition by FDA-approved ICIs, including atezolizumab, avelumab, pembrolizumab, and nivolumab, has been reported [35]. Pembrolizumab (PD-1 inhibitor) was FDA-approved as a first-line treatment for patients with unresectable or refractory MSI-H/dMMR CRC [36]. Although these monoclonal antibodies show a remarkable therapeutic effect in clinical trials, their use is limited because of immune-related adverse reactions, difficulty in manufacturing, poor delivery, restricted permeability into tumor tissues because of their large size, high production cost, and severe side effects resulting from a long biological half-life [37,38]. In contrast, small molecule inhibitors have many advantages, including high tumor penetration, favorable oral bioavailability, fewer side effects, self-administration, lower manufacturing cost, and shorter biological half-life [39]. Clinical studies indicate that small molecule inhibitors are biologically safe and can block tumor growth as effectively as PD-1/PD-L1-based ICI monotherapy [40].

In this study, we focused on the use of herbal medicines as small molecule inhibitors for tumor immunotherapy by evaluating the medicinal herb, Korean Red Ginseng, which is known to contain active components. The results indicate that RGE affects CRC through different mechanisms. Kim et al. demonstrated that RGE decreased CRC tumor cell invasion and migration as well as hypoxia-induced epithelial to mesenchymal transition by inhibiting the activation of the NF-κB and ERK1/2 signaling pathways [41]. Jeong et al. reported that RGE increased apoptosis in human CRC cells through Noxa activation by generating reactive oxygen species and endoplasmic reticulum stress [42]. Although various CRC studies have shown that RGE is useful for treating CRC, the anticancer effect of RGE on CRC through T cell activation has not been defined.

In the present study, we demonstrated that RGE treatment results in anti-CRC activity through CD8^+^ T cell-mediated antitumor immunity. First, we showed the capacity of RGE to disrupt the PD-1/PD-L1 interaction by a PD-1/PD-L1 protein binding ELISA. We found increased release of IL-2 in tumor-infiltrated CD8^+^ T cells by RGE treatment. Moreover, in a co-culture cell model, the antitumor immune effect of RGE occurred by enhancing hPD-1 tumor-infiltrated CD8^+^ T cell activity and killing hPD-L1 MC38 CRC cells with non-toxic doses of hPD-1^+^CD8^+^ T cells. We also confirmed that tumor-infiltrating CD8^+^ T cells activated by RGE release GrB to kill MC38 cells. We further examined the antitumor effect of RGE by establishing humanized PD-1/PD-L1 knock-in mice tumor-bearing MC38 allografts. The results demonstrated that RGE and Keytruda (anti-PD-1 antibodies) markedly suppressed MC38 tumor growth. RGE and Keytruda not only significantly enhanced CD8^+^ T cell infiltration into tumor tissues, but released GrB produced by CD8^+^ T cells into the tumor microenvironment. Taken together, RGE enhances the potential antitumor immunogenic T cell response by modulating the PD-1/PD-L1 axis in CRC.

Ginsenosides, a group of dammarane triterpenoids, include ginsenoside Rg3, ginsenoside Rh2, ginsenoside Rh4, and ginsenoside Rk1. They are derived from RGE and these active compounds modulate PD-L1 expression, and thus, exhibit immunomodulatory potential against immune checkpoints [39]. Ginsenoside Rg3 attenuates PD-L1 expression in cisplatin-resistant human lung cancer cells [43]. Ginsenoside Rh4 inhibits aerobic glycolysis and inhibits PD-L1 expression through the AKT/mTOR pathway in esophageal cancer [44]. Ginsenoside Rk1 downregulate PD-L1 expression in human lung cancer cells by inhibiting the NF-κB pathway in an A549 xenograft mouse model [45]. In addition, Ginsenoside Rh2 suppressed melanoma tumor growth and increased the survival in a B16F10 tumor model by enhancing CD4^+^ and CD8^+^ T cell infiltration into the tumor tissue [46]. In the present study, we confirmed that the content of ginsenoside Rg3, the predominant compound, was 6 mg/mL, confirming that it was in high abundance in RGE. In our previous study, among 12 ginsenosides, we found that ginsenoside Rg3, Rh2, and compound K were potent immune checkpoint inhibitors of human PD-1/PD-L1 binding axis by in vitro competitive ELISAs and in silico protein-ligand docking simulations [30]. Thus, ginsenosides in RGE may activate anticancer T cell immunity by targeting immune checkpoints and modulating the tumor microenvironment to improve the immune response, leading to CRC cell death. RGE has been prescribed in Korea for the treatment of cancer patients. We demonstrated the immunological effects of RGE, rather than individual ginsenosides, through PD-1/PD-L1 blockade, thus providing further evidence for its use as a cancer treatment.

## 4. Materials and Methods

### 4.1. Preparation of RGE

Korean Red Ginseng was provided by the JADAM Co. (Yangju, Korea). The dried rootlet of Korean Red Ginseng (100 kg) was mixed with 800 L of 70% ethanol for 12 h. After immersion, the RG was extracted with 70% ethanol at 60 °C for 12 h using an electric extractor (Daesung, Seoul, Korea) and the extract was filtered through a 100-mesh sieve. After filtration, the RG was condensed at 60° using a rotary evaporator (Daesung, Seoul, Korea). The free-dried extract powder (30 kg; yield 30%; abbreviated as RGE) was dissolved in water and analyzed for Rg3 content in RGE or used for experiments.

### 4.2. PD-1/PD-L1 Protein Interaction Assay

The effect of RGE on protein-protein interactions including human PD-1-PD-L1 were analyzed by a competitive ELISA (#72005, BPS Bioscience, San Diego, CA, USA). Briefly, the ligand of recombinant human PD-L1 (#71104, BPS Bioscience) was coated (100 μL, 1 μg/mL) onto a 96-well plate at 4 °C overnight. RGE, an anti-PD-1 antibody (#71120, BPS Bioscience), or anti-PD-L1 antibody (#71213, BPS Bioscience) was added to the coated plate followed by the addition of the biotinylated receptor protein of human PD-L1 (#71109, BPS Bioscience) at 0.5 μg/mL at room temperature (RT) and incubated for 2 h. HRP-conjugated streptavidin (#554066, BD Biosciences, San Jose, CA, USA) was added at 0.2 μg/mL at RT for 2 h. Relative luminescence was quantified using a SpectraMax L microplate reader (Molecular Devices, San Jose, CA, USA).

### 4.3. Human PD-L1 MC38 Cell Line

The human PD-L1 MC38 CRC cell line was obtained from Shanghai Model Organisms Center, Inc. (Shanghai, China). C57BL/6 murine hPD-L1 MC38 cells were cultivated in Dulbecco’s Modified Eagle Medium containing 10% (*v*/*v*) fetal bovine serum (FBS), 50 μg/mL hygromycin B, and antibiotics (100 U/mL penicillin and 100 μg/mL streptomycin), and maintained in a humidified atmosphere at 37 °C with 5% CO_2_. The cell culture solutions were obtained from Hyclone Laboratories, Inc. (GE Healthcare Life Sciences, Chicago, IL, USA).

### 4.4. Humanized PD-1/PD-L1 Knock-In Tumor Model

All animal experiments were conducted in accordance with the Care and Use of Laboratory Animals of the National Institutes of Health of Korea and were approved by the Institutional Animal Care and Use Committee of Korea Institute of Oriental Medicine (KIOM) (approval number KIOM-D-21-091). Humanized PD-1 C57BL/6J mice were obtained from Shanghai Model Organisms Center and bred under specific pathogen-free conditions at the KIOM for the duration of the experiments. To generate the humanized PD-1/PD-L1 MC38 CRC tumor allograft mouse model, 3 × 10^5^ hPD-L1 MC38 cells in 200 μL of phosphate-buffered saline (PBS) were injected subcutaneously into the right flank dorsal skin of each mouse. Tumor growth was monitored and the diameter was determined with digital calipers (Hi-Tech Diamond, Westmont, IL, USA).

### 4.5. Tumor-Infiltrating CD8^+^ T Cell Isolation and Stimulation

Tumor-infiltrated CD8^+^ T cells were isolated from hPD-L1 MC38 tumor tissues of humanized PD-1 mice. To obtain single cell suspensions, tumor tissues were minced with scissors and digested in Roswell Park Memorial Institute (RPMI) 1640 medium containing 0.5 mg/mL type IV collagenase (Sigma-Aldrich, St. Louis, MO, USA), 10% (*v*/*v*) FBS, and 1% antibiotics at 37 °C for 1 h in an incubator containing 5% CO_2_. The digestion was quenched with excess media and the samples were filtered through a 100-μm, 70-μm, and 40-μm mesh cell strainer (SPL Life Sciences, Pocheon, Korea). The tumor-infiltrated CD8^+^ T cells were purified by immunomagnetic negative selection (#19853, STEMCELL Technologies, Inc., Vancouver, BC, Canada). The CD8^+^ T cells were cultured in RPMI 1640 medium containing 10% (*v*/*v*) FBS and 1% antibiotics at 37 °C in a humidified atmosphere containing 5% CO_2_. The T cell receptors were then stimulated using Mouse CD3/CD28 T Cell Activator (Life Technologies, Carlsbad, CA, USA) at 37 °C for 72 h.

### 4.6. Cell Viability Assay

Cell viability was assessed using the methylthiazolyldiphenyl-tetrazolium bromide (MTT, Sigma-Aldrich) assay. Briefly, the cells were plated in 96-well flat bottom tissue culture plates at a density of 5 × 10^3^ cells/well and cultured with the indicated concentrations of RGE at 37 °C for 72 h. MTT reagent (0.5 mg/mL) was added and the plates were incubated in the dark at 37 °C for 4 h. The formazan was dissolved by adding DMSO and the optical density was quantified at 540 nm using a SpectraMax i3 microplate reader (Molecular Devices, San Jose, CA, USA).

### 4.7. IL-2 Measurement Assay

IL-2 released by activated tumor-infiltrating CD8^+^ T cells was assessed by a sandwich ELISA (#555148, BD Biosciences, San Diego, CA, USA). Briefly, an anti-mouse IL-2 antibody was coated onto 96-well plates (#3590, Corning, New York, NY, USA) with 0.1 M sodium carbonate (pH 9.5). Biotin antibody and streptavidin-HRP conjugate were added to each well and incubated at RT for 1 h. The relative absorbance was determined using a SpectraMax i3 microplate reader at 450 nm.

### 4.8. Co-Culture Experiments with Tumor-Infiltrated CD8^+^ T Cells and MC38 Cells

For co-culture experiments, hPD-L1 MC38 cells (5 × 10^4^ cells/1.9 cm^2^) as target cells were co-cultured with activated tumor-infiltrated CD8^+^ T cells (2.5 × 10^5^ cells/1.9 cm^2^) as effector cells at a target: effector cell ratio of 1:5 and treated with RGE (100–300 μg/mL) at 37 °C for 72 h. After co-culture for 72 h, the co-cultured hPD-L1 MC38 cell viability was measured using the cell counting kit-8 assay (Dojindo Molecular Technologies, Inc., Rockville, MD, USA). A lactate dehydrogenase (LDH) assay kit (#ab65393, Abcam, Cambridge, UK) was used to measure the cytotoxic activity on target cells via effector cells according to the manufacturer’s instructions. The absorbance of the formazan products was determined at 450 nm using a SpectraMax i3 microplate reader.

### 4.9. Granzyme B Measurement Assay

Measurement of mouse GrB released by cytotoxic CD8^+^ T cells in the co-culture supernatants was quantified by a sandwich ELISA (#88-8022, Thermo Fisher Scientific, Waltham, MA, USA). Briefly, an anti-mouse GrB antibody was coated onto 96-well plates (#9018, Corning) with PBS. The biotin antibody and Av-HRP were added to each well and incubated at RT for 1 h. The relative absorbance was determined using a SpectraMax i3 microplate reader at 450 nm.

### 4.10. In Vivo RGE Treatment

The in vivo antitumor effect of RGE was evaluated using the hPD-L1 MC38 CRC model. Male tumor-bearing humanized PD-1 knock-in mice were randomly distributed into four groups (*n* = 5 per group) and RGE treatments were initiated when the tumor volume reached 100 mm^3^ (day 14 post inoculation). Mice were injected with PBS (vehicle) or RGE (100, 300 mg/kg) daily by oral administration or Keytruda (anti-PD-1 human antibody, 2.5 mg/kg) twice per week by intraperitoneal injection for 21 days. Mice were euthanized after day 22. Tumor volume was calculated according to the formula V_t_ = (length × width^2^)/2. Tumor suppression rate was calculated with the formula (V_c_ − V_t_)/V_c_ × 100%, where V_c_ (vehicle) and V_t_ (other treatment groups) are the tumor volumes.

### 4.11. Immunohistochemistry

The tumor tissues were harvested, formalin-fixed, and paraffin-embedded. The paraffin sections were immunostained with a primary antibody against CD8 (#98941, Cell Signaling Technology, Danvers, MA, USA), GrB (#46890, Cell Signaling Technology), and CD4 (#ab183685, Abcam) using the DAKO EnVision kit (#K5007, DAKO, Jena, Germany). The sections were counterstained with Mayer’s hematoxylin. For image observation, an Olympus BX53 microscope and an XC10 microscopic digital camera (Tokyo, Japan) were used.

### 4.12. Sample Preparation for HPLC Analysis

Standards (ginsenoside Rg3, Faces Biochemical Co., Wuhan, China) were dissolved in methanol. A stock solution was prepared at a concentration of 1 mg/mL and then serially diluted with methanol (1000, 500, 250, 125, and 62.5 µg/mL). RGE was diluted with water (500, 250, 125, 62.5, and 31.25 mg/mL). All solutions for were filtered through 0.22 μm RC membrane syringe filters (Whatman, Maidstone, UK).

### 4.13. Optimization of Chromatographic Conditions

HPLC was performed using a Thermo Scientific Ultimate 3000 system (Thermo Scientific) equipped with a binary pump, auto-sampler, column oven, and diode array UV/Vis detector (DAD). Separation was carried out on a Waters Xbridge^®^ C18 column (250 × 4.60 mm, 5 μm), with an oven temperature of 40 °C and a UV wavelength of 203 nm. The mobile phase consisted of 0.1% trifluoroacetic acid (*v*/*v*) in water (solvent A) and acetonitrile (solvent B) with gradient elution of 0–60 min, 10% B (0–3 min), 10–45% B (3–20 min), and 45–80% B (20–60 min) at a flow rate of 1.0 mL/min (injection volume: 10 μL).

### 4.14. Statistical Analysis

Data are presented as the mean ± standard deviation (SD) and were performed using GraphPad Prism software (GraphPad Software, Inc., La Jolla, CA, USA). Statistical comparisons were done using a one-way analysis of variance, followed by Tukey’s post hoc test, which was used for comparisons between multiple groups, as indicated. *p*-values < 0.05 were considered statistically significant. All experiments, except animal studies, were conducted in at least three independent cases.

## 5. Conclusions

We discovered that RGE inhibits PD-1/PD-L1 interaction by in vitro protein binding ELISA and suppresses cell growth of CRC by increasing tumor-infiltrated CD8^+^ T cells immune function in co-culture cell model. In addition, we established that RGE effectively reduced hPD-L1 MC38 tumor growth via inducing the CD8^+^ T cell infiltration and GrB release in the tumor microenvironment in allograft tumor humanized PD-1/PD-L1 knock-in mouse models. Based on our results, RGE, as a potential antitumor drug with blockade of the immune checkpoint, PD-1/PD-L1 axis, warrants further preclinical investigation in patients with CRC as a potential cancer immunotherapy.

## Figures and Tables

**Figure 1 ijms-24-01894-f001:**
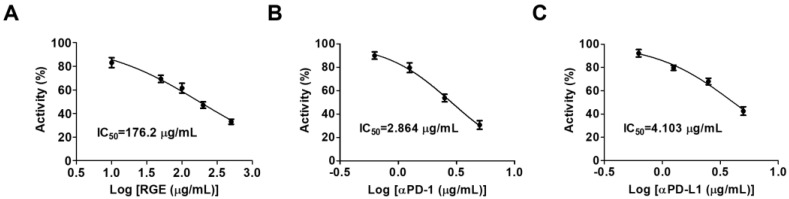
RGE blocks the human PD-1/PD-L1 interaction activity in vitro. (**A**) RGE blocks the hPD-1/hPD-L1 interaction as determined by competitive ELISA. (**B**,**C**) Human PD-1 neutralizing antibody (**B**) and human PD-L1 neutralizing antibody (**C**) act as positive controls. The assays were performed in triplicate at each concentration.

**Figure 2 ijms-24-01894-f002:**
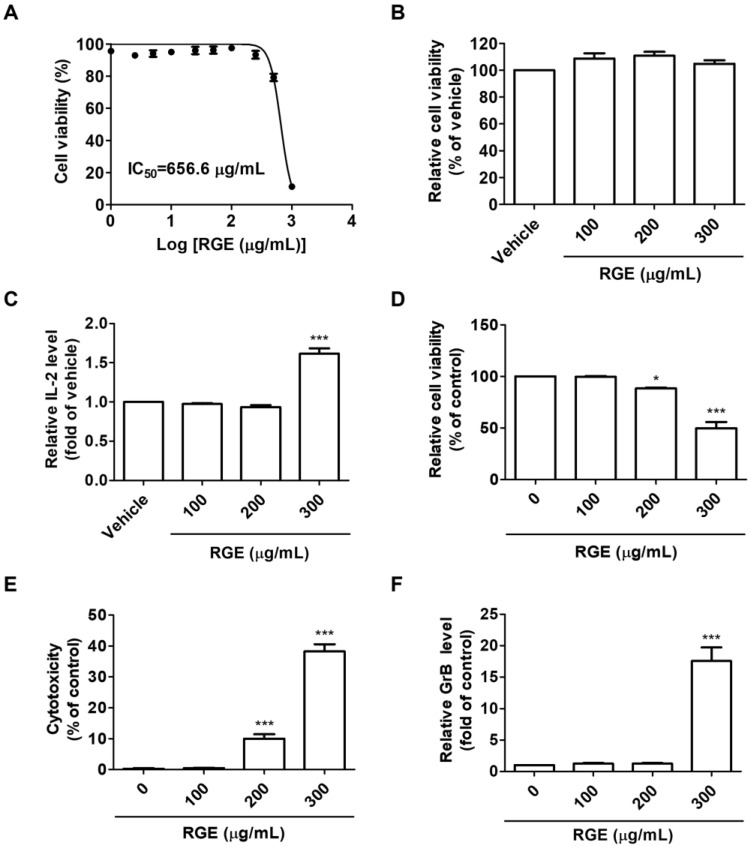
RGE activates hPD-1 tumor-infiltrating CD8^+^ T cells and cytotoxic effect of CD8^+^ T cells on hPD-L1 MC38 cancer cells. (**A**,**B**) Cell viability against hPD-L1 MC38 cells (**A**) and hPD-1 tumor-infiltrating CD8^+^ cells (**B**) conducted by MTT assay after treatment with RGE for 72 h. (**C**) Relative IL-2 levels in the supernatant of hPD-1 tumor-infiltrating CD8^+^ cells treated with RGE for 72 h as measured by a mouse IL-2 ELISA. (**D**) Co-cultured hPD-L1 MC38 cell viability was evaluated using the CCK assay. (**E**) Cytotoxicity in co-cultured hPD-L1 MC38 cells was detected by an LDH cytotoxicity assay. (**F**) Relative GrB levels in the co-culture supernatants at 72 h were determined by a mouse GrB ELISA. Data are presented as the mean ± SD, * *p* < 0.05, *** *p* < 0.001 compared with the control.

**Figure 3 ijms-24-01894-f003:**
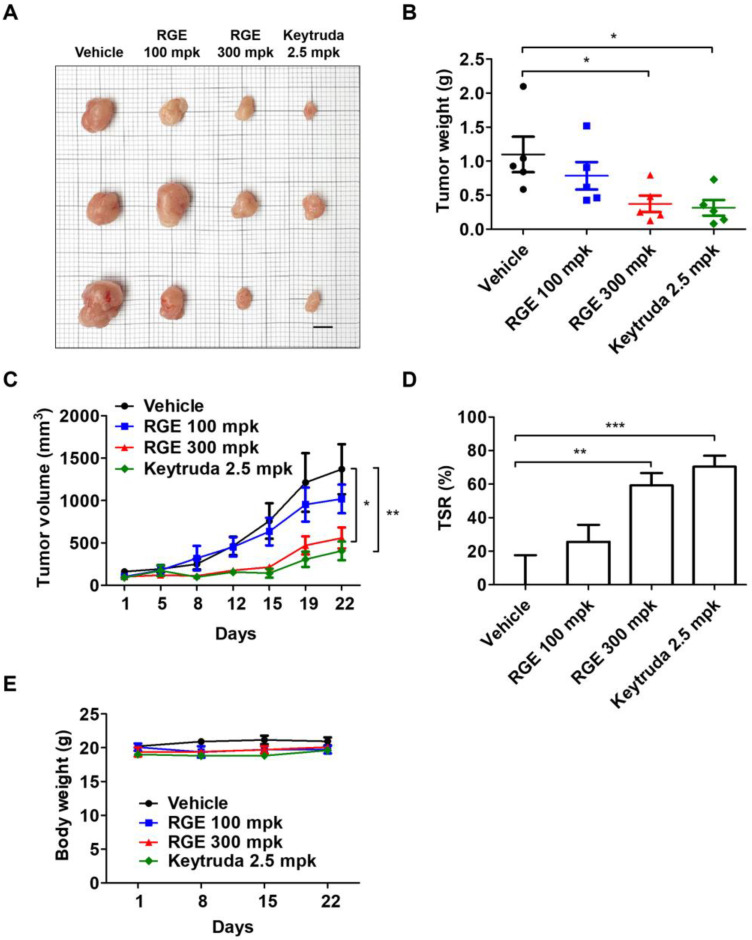
Antitumor effect of RGE in a human PD-1/PD-L1 MC38 cell allograft mouse model. (**A**) Images of the harvested tumor tissues in vehicle, RGE 100, 300 mg/kg, and Keytruda (anti-PD-1) 2.5 mg/kg groups on day 22 (n = 3, bar indicates 1 cm). (**B**) Tumor weight on day 22. (**C**) The growth curve based on tumor volume over 22 days. (**D**) Tumor suppression rate on day 22. (**E**) Body weight (grams) over 22 days. Data are presented as the mean ± SD, * *p* < 0.05, ** *p* < 0.01, *** *p* < 0.001 compared with the control.

**Figure 4 ijms-24-01894-f004:**
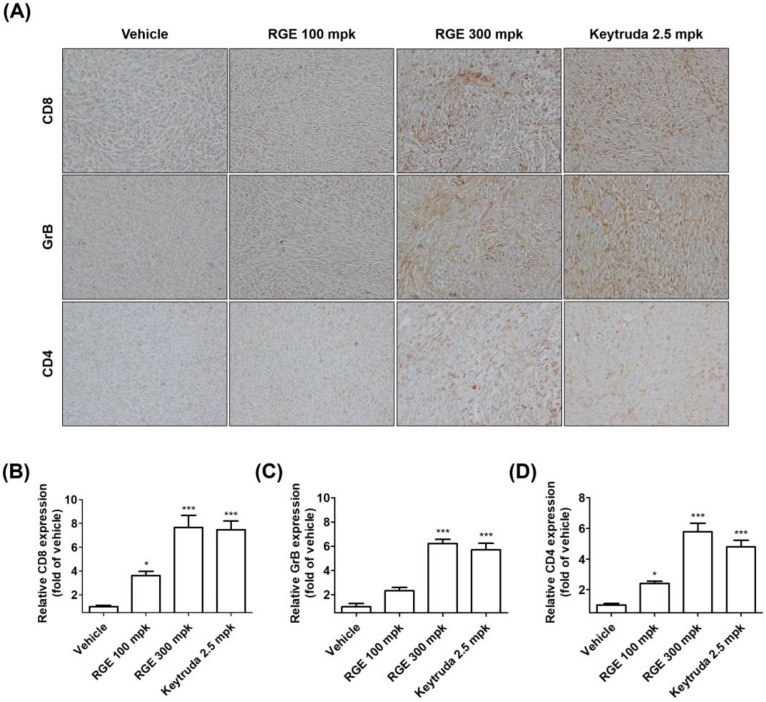
RGE activates hPD-1 tumor-infiltrating CD8^+^ T cells in vivo by immunohistochemical staining. (**A**) Representative images (×400) of CD8^+^, GrB, and CD4^+^ staining of tumor tissues by immunohistochemical analysis. (**B**–**D**) Quantification of CD8^+^ (**B**), GrB (**C**), and CD4^+^ (**D**) expression in tumor tissues. Data are presented as the mean ± SD, * *p* < 0.05, *** *p* < 0.001 compared with the control.

**Figure 5 ijms-24-01894-f005:**
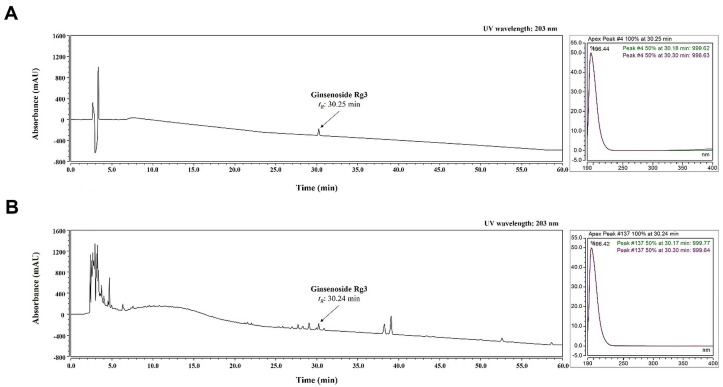
HPLC analysis of ginsenoside Rg3 in RGE at UV wavelengths of 203 nm. UV chromatograms of standards of ginsenoside Rg3 (**A**) and ginsenoside Rg3 in RGE (**B**).

**Table 1 ijms-24-01894-t001:** Regression data and content of ginsenoside Rg3 in RGE.

Analytes	Regression Equation	*R^2^*	Content (mg/g)
ginsenoside Rg3	y = 0.0702x + 0.5987	0.9999	6.0423 ± 0.0228

## Data Availability

The original contributions presented in the study are included in the article. Further inquiries can be directed to the corresponding author.

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
