# Peer review of "Antitumor Effect of Korean Red Ginseng through Blockade of PD-1/PD-L1 Interaction in a Humanized PD-L1 Knock-In MC38 Cancer Mouse Model"

_ijms, 2023, doi:10.3390/ijms24031894_

Round 1
Reviewer 1 Report
Comments:
In this manuscript, the authors described “Antitumor Effect of Korean Red Ginseng through Blockade of PD-1/PD-L1 Interaction in a Humanized PD-L1 Knock-In MC38 Cancer Mouse Model”. This paper show that RGE as a potential antitumor drug with blockade of immune checkpoint, PD-1/PD-L1 axis. However, there are a few points that need to be clarified.
Comment
1. The content of ginsenoside was greatly increased by RGE, thus ginsenoside Rg3 was selected as a standard for analysis. Rg3 and compound K were identified for their interaction efficacy with PD-1/PD-L1, which supported the blocking activity against PD-1/PD-L1 binding interactions (1). The author shall be identified the amount of compound K.
Reference:
1. Yim, N. H.; Kim, Y. S.; Chung, H. S., Inhibition of Programmed Death Receptor-1/Programmed Death Ligand-1 Interactions by Ginsenoside Metabolites. Molecules 2020, 25, (9), 2068.
2. When T cells are activated, the quantity of CD8+ cytotoxic lymphocytes are increased and cytokines, such as GZMB and IFN-γ, are secreted into the tumor microenvironment to initiate killing of the tumor cells. The author shall be identified GZMB, and IFN-γ protein levels in the immunohistochemical staining.
3. RGE enhances the abundance of CD8+ T cell infiltration on hPD- L1 MC38 tumors. Release of IFNγ by CD8+ T cells with DCs was increased by the presence of Keytruda. The author shall be identified CD4+ expressions.
Reviewer 2 Report
In this manuscript, Eun-Ji Lee et al showed that anti-tumor effect of Korean Red Ginseng through blockade of PD-1/PD-L1 interaction in a humanized PD-L1 knock-in MC38 cancer mouse model. These findings are potentially interesting. The manuscript could be further strengthened with a few additional experiments denoted below.
1. Authors need to explain more about function of Korean Red Ginseng in introduction part.
2. It would be more significant if biological relevance of the regulation of PD-1 by Korean red ginseng in cancer cell development can be addressed using protein samples, such as by western blotting.
3. Significance notation created in figure legend cannot be found in figure1.
Round 2
Reviewer 1 Report
accepted